# Evaluating Ammonia Toxicity and Growth Kinetics of Four Different Microalgae Species

**DOI:** 10.3390/microorganisms12081542

**Published:** 2024-07-27

**Authors:** Umut Metin, Mahmut Altınbaş

**Affiliations:** 1Department of Environmental Engineering, Faculty of Civil Engineering, Yıldız Technical University, 34220 Istanbul, Turkey; umetin@yildiz.edu.tr; 2Department of Environmental Engineering, Faculty of Civil Engineering, Istanbul Technical University, 34469 Istanbul, Turkey

**Keywords:** microalgae, ammonia toxicity, nitrogen, wastewater

## Abstract

Although wastewater with high ammonia concentration is an ideal alternative environment for microalgae cultivation, high ammonia concentrations are toxic to microalgae and inhibit microalgae growth. In this study, the ammonia responses of four widely used microalgae species were investigated. *Chlorella vulgaris*, *Chlorella minutissima*, *Chlamydomonas reinhardtii* and *Arthrospira platensis* were grown in batch reactors maintained at seven different NH_4_Cl concentrations at a constant pH of 8. Growth and nitrogen removal kinetics were monitored. IC_50_ values for the mentioned species were found as 34.82 mg-FA/L, 30.17 mg-FA/L, 27.2 mg-FA/L and 44.44 mg-FA/L, respectively, while specific growth rates for different ammonia concentrations ranged between 0.148 and 1.271 d^−1^. *C. vulgaris* demonstrated the highest biomass growth under an ammonia concentration of 1700.95 mg/L. The highest removal of nitrogen was observed for *A. platensis* with an efficiency of 99.1%. The results showed that all tested species could grow without inhibition in ammonia levels comparable to those found in municipal wastewater. Furthermore, it has been concluded that species *C. vulgaris* and *A. platensis* can tolerate high ammonia levels similar to those found in high strength wastewaters.

## 1. Introduction

Microalgae can function as green chemical factories that can convert sunlight, carbon dioxide, and nutrients into various valuable compounds. They are significant sources of high-value products like carotenoids, long-chain polyunsaturated fatty acids, and phycobilins, as well as fuels or their precursors [1,2,3]. However, the high costs associated with cultivation make the production of many of these valuable products economically unfeasible. 

Nutrient-rich wastewater can serve as a cost-effective nutrient source for microalgae cultivation [4]. Algal wastewater treatment offers a more economical and efficient method for removing nutrients and metals from wastewater compared to conventional tertiary treatments [5]. For example, traditional phosphorus removal methods do not recycle phosphorus, as it is extracted along with various waste products, some of which are toxic [6]. Moreover, while effective for total nitrogen removal, the processes of nitrification and denitrification are costly and complex, and they transfer nitrogen to the atmosphere, rendering it unusable as a nutrient for biomass production [7]. In contrast, algae can assimilate these nutrients into their biomass, effectively removing them from wastewater [4]. Additionally, using microalgae eliminates the need for sludge treatment, which poses significant logistical challenges [8]. Furthermore, by leveraging their metabolic flexibility to grow phototrophically, heterotrophically, or mixotrophically, microalgae serve as an effective system for treating a diverse array of wastewaters [2,4].

Nitrogen is one of the essential macronutrients required for the growth of microalgae [9]. The supply of nitrogen for microalgal production constitutes one of the primary nutrient costs [10]. More than 45% of the effective energy input in microalgae cultivation is derived from nitrogen input [11]. The use of nitrogen-containing waste streams, such as effluent from secondary wastewater treatment plants, anaerobic digester centrate, dairy waste runoff, or post-lipid-extracted biomass, can serve as a nitrogen source to reduce costs associated with nitrogen demand [12]. The main nitrogen form in wastewaters is ammonium [13,14], which can be found in some wastewaters at concentrations up to 3000 mg/L [15]. Ammonium is the ideal nitrogen source for microalgal growth because it does not require a redox reaction for assimilation, thus consuming less energy. In contrast, other inorganic nitrogen forms must be reduced to ammonium within the cell [16]. Therefore, ammonium-rich wastewaters would serve as an ideal alternative culture medium for microalgae cultivation [13]. However, elevated ammonium levels have been reported to inhibit microalgal growth or even result in cell death [17,18]. Consequently, the efficiency of algal wastewater treatment does not reach the desired level.

Ammonia toxicity in water results from both unionized NH_3_ (free ammonia, FA) and ionized ammonium (
${\mathrm{NH}}_{4}^{+}$
). NH_3_ is considered the most toxic form because it is uncharged, lipid-soluble, and can easily diffuse across membranes [17,19]. In many phototrophic microorganisms, unionized ammonia can disrupt autotrophic metabolism. FA shock negatively affects genetic information processing, photosynthesis, and nutrient metabolism [20]. It causes an increase in the activation of cell cycle checkpoints, programmed apoptosis, and the activity of related genes, leading to microalgal growth inhibition, inhibition of photosynthetic activities (including both light and dark reactions) and photoprotection through non-photochemical quenching [20]. NH_3_ can bind to the core of the photosystem II oxygen evolution reaction, increasing photosensitivity and ultimately causing greater photosystem damage [12,17]. Additionally, it can uncouple photophosphorylation by diminishing the pH gradient necessary for converting ADP to ATP [12]. It has also been suggested that oxidative stress induced by FA could affect the activities of antioxidative enzymes and lead to cellular disorders [19]. Currently, there is no chemical method to separately measure these two forms of ammonia. Existing methods measure them together, commonly referred to as ‘total ammonia’ (
${\mathrm{NH}}_{4}^{+}$
 + NH_3_). The relative concentration of each form is strongly dependent on pH, while temperature and salinity have minor effects [17].

The amounts of available nitrogen in the culture directly influence the growth of microalgae [21]. Optimizing nitrogen supply is critically important for microalgal productivity [11]. While there is an extensive body of literature on microalgae growth at different ammonia concentrations, cellular response to different ammonia concentrations is highly dependent on abiotic conditions such as pH, temperature, light intensity and salinity [12,17] which adds complexity to the assessment of ammonia tolerance of species. Furthermore, the lack of fully specified conditions in some studies complicates comparisons even more. Moreover, each microalgae species responds differently to different ammonia concentrations [12,17]. Ammonia toxicity resulting from high ammonium concentration creates a significant technical bottleneck in the biological treatment of wastewater [22]. For cost-effective microalgae production and the treatment of wastewater with high ammonia concentrations, it is crucial to utilize species that exhibit high tolerance to ammonia under adequate abiotic conditions.

When the literature was examined, three green algae species *Chlorella vulgaris*, *Chlorella minutissima* and *Chlymodomanas reinhardtii* and the cyanobacteria species *Arthrospira platensis* were selected to be used in this study, taking into account their tolerance to high ammonia concentration, growth capacity, ability to produce valuable products and competence in wastewater treatment [9,13,17,23,24,25,26]. 

These species have been frequently used in the treatment of wastewater containing high amounts of ammonia. *C. vulgaris* has been cultivated in various types of wastewater, including effluents from the gelatine industry, anaerobic digestion of poultry litter, and piggery wastewater [27,28,29]. It has been observed that *C. vulgaris* can grow in piggery wastewater containing 678 mg-N/L ammonia [27]. However, in these studies, it was observed that *C. vulgaris* reproduced at a maximum FA concentration of around 20 mg/L [29]. The ammonia concentrations in studies involving *C. minutissima* are lower compared to those for *C. vulgaris*. It has been observed that *C. minutissima* is used for the treatment of wastewater with ammonia concentrations of up to approximately 250 mg-N/L [30,31]. Compared to the other species used in this study, there are fewer studies on the cultivation of *C. reinhardtii* in wastewater. Examples of this wastewater are wastewater treatment plant effluent and piggery wastewater [25,26]. However, in these studies, wastewater with low ammonia concentrations (67 mg-N/L) was used [26]. *A. platensis* has been cultivated in various types of wastewater such as lac wastewater, swine wastewater treatment effluent, cassava processing wastewater and wastewater effluent from the petrochemical industry [28,32,33,34]. It has been observed that *A. platensis* can grow in anaerobically digested cattle wastewater with an ammonia concentration of 336 mg-N/L [35].

The mentioned species were cultivated under 
${\mathrm{NH}}_{4}^{+}$
 concentrations ranging from 25 to 1000 mg/L. The growth conditions were selected based on the anticipated environmental conditions (temperature, pH, and light intensity) for treating wastewater with microalgae. A detailed explanation is provided in Section 2. Growth and inhibition kinetics were determined for each case to measure the effect of ammonia toxicity. Also, nitrogen removal kinetics were investigated in each case.

Although numerous studies have demonstrated the potential of various microalgae in treating different types of wastewater, a universally accepted, cost-effective, and efficient method has yet to be established [22]. This study provides a comprehensive analysis of the ammonia responses of four widely studied microalgae species, focusing on their growth and nitrogen removal capabilities. Consequently, the microalgae species with the highest biomass production efficiency and nitrogen removal rate under defined conditions suitable for wastewater or systems with high ammonia content have been identified, establishing a foundation for future research involving the direct application of wastewater. 

## 2. Methods

### 2.1. Species and Growth Conditions

Axenic unialgal cultures of *Chlorella vulgaris* (CCAP 211/116) and *Arthrospira platensis* (SAG 85.79) species were purchased from Phycotec Biyoteknoloji A.Ş., while, *Chlorella minutissima* (ELSTER 1998/9) and *Chlamydomonas reinhardtii* (SMITH 1980/UTEX 2246) species were obtained from the Culture Collection of Autotrophic Organisms (CCALA). 

Seed cultures were grown in Bold’s Basal Media (BBM) until they reached the early-stationary phase in order to achieve high cell density. The composition of the BBM is given in Table 1. The experimental cultures were inoculated with seed culture to achieve an initial optical density of around 0.1–0.2 at 680 nm. It was ensured that the volume used for inoculation was always less than 1% of the total working volume. Seed cultures were adapted to the environmental conditions by making two batch sets before inoculating the experimental cultures. All conditions during growth for the seed cultures were the same as for the experimental cultures except for nitrogen concentration. For all the growth experiments, seed cultures were cultivated using BBM containing ammonium chloride in place of sodium nitrate to maintain an equivalent nitrogen content. Under the experimental conditions, the free ammonia (FA) concentration in the seed cultures was calculated as 3.58 mg/L.

Growth experiments were carried out in 250 mL Erlenmeyer flask batch reactors with a working volume of 100 mL. Microalgae were cultivated with ambient air. The air was allowed to enter the culture medium in the flasks by passing through the nitrocellulose filter with a pore diameter of 0.45 μm, which closed the mouth of the flasks. The reactors were not shaken during the growth experiments.

Growth conditions were determined by taking into account the optimal conditions achievable during wastewater treatment under natural conditions in our region (Istanbul, Turkey). In general, a light intensity of 200–400 µmol photons/m^2^/s is considered the light saturation level for most microalgae species [36], typically reached early in the day during spring and summer [37]. Therefore, close to the average of the given range, microalgae were grown under cold white LED light with a photosynthetic flux density of 280 ± 10 μmol photon/m^2^/s by continuous illumination (MQ-200X Quantum Separate Sensor with Handheld Meter, Apogee Instruments, North Logan, UT, USA). The average highest temperature in Istanbul during the summer months varies between 26 and 30 °C [38]. Therefore, close to the average of the given range, the temperature was kept constant at 28 °C during growth. The pH was adjusted to be around 8 during growth. This pH level was selected to minimize ammonia stripping during growth and was further validated through an ammonia stripping control experiment [39,40].

Experimental cultures were grown in 
${NO}_{3}^{-2}$
 omitted BMM medium. Each experimental run lasted until the early/mid-stationary phase. Ammonium chloride was used as the ammoniacal nitrogen source. Seven different ammonium concentrations were used to determine ammonia toxicity. The concentrations used in the toxicity test are 25, 50, 100, 250, 500, 750 and 1000 mg-
${\mathrm{NH}}_{4}^{+}$
/L. All different concentrations were performed in biological triplicates. To control the concentration of FA, the experimental media contained 10 mM HEPES buffering agent adjusted to a pH of 8. At the beginning of growth at 28 °C and pH 8, FA concentrations in the media were 1.69, 3.39, 6.77, 16.93, 33.85, 50.78 and 67.70 mg-FA/L, respectively. FA concentration was calculated according to the equation given by Cao et. al. [41]. Equation (1) used to calculate the amount of FA is given below.

(1)
FA(mg/L)=1814×NH4+—N×10pH10pH+e(6334273+t)


In this equation, t is defined as ambient temperature (°C). 

The pH value was 8 at the beginning of growth, and the pH remained in the range of 8 ± 0.1 during the growth period. A constant pH was maintained by using sterile 1 N H_2_SO_4_ and 1 N NaOH, when necessary. No nitrogen was added to the media during growth.

### 2.2. Sampling and Analysis

Optical density measurements were performed daily. Optical density measurements were performed at a wavelength of 680 nm (OD_680_) using a spectrophotometer (BIO-Rad SmartSpec Plus, Hercules, CA, USA). 

The pH of the samples was measured daily using a pH meter ensuring the cultures remained sterile.

Concentrations of ammonium were measured using the standard method 4500-NH_3_/F Phenate Method without major modification [42]. The modification implemented involves proportionally reducing all used volumes to accommodate working with smaller quantities. When measuring ammonia, spectrophotometric measurements were made using BIO-Rad SmartSpec Plus. At the end of the growth, total organic nitrogen measurement using standard method 4500-N_org_ B. Macro—Kjeldahl Method without modification [42].

The amount of nitrogen stripping into the air in the ammonia stripping control experiment was determined by subtracting the final nitrogen amount from the initial nitrogen amount.

(2)
$$\mathrm{Nitrogen}\;\mathrm{stripping}\;\left(\mathrm{mg}/L\right)={{\mathrm{NH}}_{4}^{+}—N}_{\mathrm{initial}}-{{\mathrm{NH}}_{4}^{+}—N}_{\mathrm{final}}$$


### 2.3. Biomass Concentration Estimation and Measurement

The amount of dry biomass was measured using the standard method 2540D—Total Solids Method Dried at 103–105 °C using a Millipore type AP40047 (MERCK, Darmstadt, Germany) filter [42]. Under the conditions studied, since the media do not contain inorganic suspended solids, the total amount of solids is equal to the amount of organic matter, which is the dry biomass weight (DBW). DBW measurements were performed to find the relationship between the optical density (OD_680_) and dry biomass of the microalgae species examined in the study and to measure the total biomass obtained at the end of the growth studies.

For the estimation of the dry weight of biomass, the calibration curve between dry cell weight and absorbance value at 680 nm was plotted. To obtain the optical density (OD_680_)—dry biomass weight (mg/L) curves, the studied cultures in the late exponential phase-early stationary phase were properly diluted with fresh modified BBM. For each dilution, the optical density and DBW were measured. The obtained values were used to draw the calibration curve. The estimation of dry biomass concertation was calculated based on the obtained respective calibration equations. These equations are given below:
(3)
$$C.\;\mathrm{vulgaris}:\;\mathrm{DBW}\left(\mathrm{mg}/L\right)=185.52\;\times \;{\mathrm{OD}}_{680};\;R^{2}=0.986$$


(4)
$$C.\;\mathrm{minutissima}:\;\mathrm{DBW}\left(\mathrm{mg}/L\right)=188.72\;\times \;{\mathrm{OD}}_{680};\;R^{2}=0.994$$


(5)
$$C.\;\mathrm{reinhardtii}:\;\mathrm{DBW}\left(\mathrm{mg}/L\right)=167.51\;\times \;{\mathrm{OD}}_{680};\;R^{2}=0.990$$


(6)
$$A.\;\mathrm{platensis}:\;\mathrm{DBW}\left(\mathrm{mg}/L\right)=171.56\;\times \;{\mathrm{OD}}_{680};\;R^{2}=0.999$$


### 2.4. Growth Kinetics Modeling

To find the growth rates of the investigated algal species, the modified Gompertz model, refined for modeling biological systems into a more convenient non-linear growth model, was employed to characterize the growth kinetics under the investigated conditions [43,44]. It has been observed that the modified Gompertz model explains the growth of microalgae better than other growth models [45,46]. Equation (7) of the modified Gompertz model is given below.

(7)
$$\ln\left(\mathrm{DBW}\right)=\ln\left({\mathrm{DBW}}_{0}\right)+\ln(\frac{{\mathrm{DBW}}_{\max}}{{\mathrm{DBW}}_{0}})\;\times \;e^{{-e}^{\left[\frac{e\;\times \;\mu }{\ln(\frac{{\mathrm{DBW}}_{\max}}{{\mathrm{DBW}}_{0}})}*\left(t_{\mathrm{lag}}-t\right)\;+\;1\right]}}$$


In this given equation, DBW is the dry biomass concentration at given time (mg/L), DBW_0_ is the calculated initial dry biomass concentration (mg/L), DBW_max_ is calculated maximum dry biomass concentration (mg/L), μ is the highest specific growth rate achieved during growth (time^−1^), t_lag_ is the length of the lag phase (hours), and t is the sampling time (hours).

Estimated biomass concentration data calculated using OD_680_ values obtained from growth experiments were used in growth kinetics modeling. The fitting procedure was performed using commercial computer software Microsoft Excel 2016 and unknown parameters were determined by non-linear regression analysis. Under the defined conditions, the DBW_0_, DBW_max_, μ and t_lag_ parameters were optimized for the best fit by the non-linear GRG method.

The doubling time was calculated using the specific growth rate obtained from the modified Gompertz model with the help of Equation (8) given below.

(8)
$$t_{d}=\frac{\ln(2)}{\mu }$$


In this given equations t_d_ is the doubling time.

### 2.5. Inhibition Kinetics Modeling

Toxicity was expressed as an IC_50_ value. It refers to the inhibitory concentration that corresponds to a 50% reduction in cell growth rate [12]. Specific growth rate data obtained from the growth kinetics model were used for substrate inhibition modeling to calculate the IC_50_ value.

The Monod-based substrate inhibition model developed by Han and Levenspiel was employed to assess the inhibition kinetics [47]. Equation (9) is given below:
(9)
$$\mu ={\mu_{\max}\left(1-\frac{C_{s}}{k_{i}}\right)}^{n}\times \frac{C_{s}}{C_{s}+k_{m}{\left(1-\frac{C_{s}}{k_{i}}\right)}^{m}}$$


In these given equations μ_max_ is the maximum specific growth rate (time^−1^), C_s_ is the substrate concentration (mg/L), k_i_ is the inhibitor concentration (mg/L), k_m_ is the monod half-saturation concentration of growth kinetics (mg/L), and m and n are unitless constants that describe observed changes to μ_max_ and k_m_.

The fitting procedure was performed using commercial computer software Microsoft Excel 2016 and unknown parameters were determined by non-linear regression analysis. When performing regression analysis, the limit values for μ_max_ and k_m_ were determined using the Monod equation. The values of μ_max_ and k_m_ obtained from the Monod equation were used as the boundary parameters. The monod Equation (10) is given below.

(10)
$$\mu =\mu_{\max}\frac{C_{s}}{{k_{m}+C}_{s}}$$


Under the defined conditions, the μ_max_, m, n, K_m_ and K_i_ parameters were optimized for the best fit by the non-linear GRG method.

### 2.6. Determination of Nitrogen Removal Parameters

The amount of nitrogen in the biomass obtained at the end of the experiment was determined using the following equation.

(11)
$$N.\;\mathrm{content}\;(\%)=\frac{{\mathrm{Organic}—N}_{\mathrm{end}}}{{\mathrm{DBW}}_{\mathrm{end}}}$$


In this given equation, Organic—N_end_ is the organic nitrogen concentration of biomass at the end of cultivation (mg/L), and DBW_end_ is the biomass concentration at the end of cultivation (mg/L)

Nitrogen removal efficiency was calculated with the equation below.

(12)
$$N.\;\mathrm{Removal}\;\mathrm{Efficiency}\;\left(\%\right)=(1-\frac{{{\mathrm{NH}}_{4}^{+}—N}_{\mathrm{final}}}{{{\mathrm{NH}}_{4}^{+}—N}_{\mathrm{initial}}})\times 100$$


In this given equation, 
${{\mathrm{NH}}_{4}^{+}—N}_{\mathrm{initial}}$
 is the initial ammonia nitrogen concentration (mg/L), and 
${{\mathrm{NH}}_{4}^{+}—N}_{\mathrm{final}}$
 is the final ammonia nitrogen concentration (mg/L).

Biomass yield was determined by measuring the amount of biomass produced per unit of nitrogen consumed. Efficiency calculation given below.

(13)
$$Y_{X/N}=\frac{{\mathrm{DBW}}_{\mathrm{end}}}{{{\mathrm{NH}}_{4}^{+}—N}_{\mathrm{initial}}-{{\mathrm{NH}}_{4}^{+}—N}_{\mathrm{final}}}$$


Following equations were used to calculate the average removal rates.

(14)
Ave. N. Removal Rate (mgL×day)=NH4+—Ninitial−NH4+—Nfinaltgrowth


In this given equation t_growth_ is the total growth time (day).

### 2.7. Statistics

All the growth experiments were conducted in triplicates. The resulting algal concentration data presented are shown as mean with standard error. 

One-way analysis of variance (ANOVA) was carried out to evaluate the difference in biomass production and average nitrogen removal rate under different ammonium concentrations at the confidence coefficient level of α = 0.05. Necessary calculations regarding ANOVA were performed using commercial computer software Microsoft Excel 2016.

The goodness of fit of the growth and inhibition models was evaluated through root-mean-square error (RMSE) and adjusted coefficient of determination (adj. R^2^).

The RMSE was calculated according to Equation (15) which is given below.

(15)
$$\mathrm{RMSE}=\sqrt{\frac{\sum_{i=1}^{n}{\left({\mathrm{Pd}}_{i}-{\mathrm{Ob}}_{i}\right)}^{2}}{\left(n-p\right)}}$$


In this given equation Pd_i_ is the values predicted by the model, Ob_i_ is the actual experimental data, n is the number of experimental data p is the number of parameters of the assessed model

A smaller RMSE value indicates that the model fits better with the experimental results.

Since growth and inhibition models used in work are non-linear models, adjusted R^2^ is used to calculate the quality models. Adj. R^2^ was calculated according to Equation (16) which is given below.

(16)
$$\mathrm{Adj}.\;R^{2}=1-\frac{(1-R^{2})(n-1)}{\left(n-p-1\right)}$$


The closer the resulting “adj. R^2^” value is to 1, the better the models fit the experimental results.

## 3. Results and Discussion

### 3.1. Effects of Ammonia on Growth

In this study, four different microalgae species were grown in media with concentrations ranging from 25 to 1000 mg-
${\mathrm{NH}}_{4}^{+}$
/L, which corresponds to 1.69 to 67.70 mg-FA/L at pH of 8. The ammonia concentrations tested in this study correspond to the ammonia concentrations that can be seen in municipal and industrial wastewater [13,17].

The four species examined in this study exhibited growth inhibition at different rates (Figure 1). According to the results obtained, it was observed that the specific growth rates of all microalgae tested increased with increasing FA concentration, up to a concentration of 6.77 mg-FA/L. However, it should be noted that the increase in specific growth rate is not proportional to the increasing FA concentration. It was observed that when the concentration increased to 16.93 mg-FA/L, the specific growth rate of all microalgae species decreased slightly, although at varying rates. It has been observed that a further increase in FA concentration results in a significant decline in the specific growth rate in all tested species. Moreover, none of the species could grow at a concentration higher than 67.7 mg-FA/L (1000 mg-
${\mathrm{NH}}_{4}^{+}$
/L).

Although the seed cultures used were acclimated to high ammonia concentrations, it was observed that the time spent by the experimental cultures in the lag phase increased with increasing FA concentration. While no lag phase was observed in cultures with concentrations of 1.69 and 3.39 mg-FA/L, a lag phase of 87.86 h was observed when the concentration increased to 67.7 mg-FA/L. This indicates that an extended period is necessary for the acclimatization of seed cultures, particularly when dealing with high FA concentrations. Supporting our conclusions, a study by Collos and Harrison [17] demonstrated that cells should be acclimated to these extremely high ammonium concentrations through four transfers, each spaced seven days apart.

The variables obtained by performing non-linear regression analysis from the Han and Levenspiel equation (Equation (9)) are given in Table 2. The obtained results showed that the growth of the examined microalgae is in accordance with the substrate inhibition pattern.

According to the results obtained, when *A. platensis* is excluded, even though the three green algae species examined in the experiment had similar maximum growth rates, *C. vulgaris* seems to be the species most resistant to ammonia toxicity. *C. vulgaris* showed 50% growth inhibition (IC_50_), at 34.82 mg-FA/L, which corresponds to about 515.5 mg-
${\mathrm{NH}}_{4}^{+}$
/L. This result falls within the wide range of responses of *C. vulgaris* to ammonia toxicity in the literature [13,17,48,49,50]. According to the review study by Collos and Harrison [17], an IC_50_ value for *C. vulgaris* has been reported to be as low as 53.27 mg-N/L, corresponding to an FA concentration of 3 µM. In the study conducted by Przytocka-Jusiak et al. [48], the IC_50_ value of 330 mg-N/L was reported for *C. vulgaris*. On the other hand, it has been shown that *C. vulgaris* can grow at much higher ammonia concentrations than the ammonia values observed in this study. *C. vulgaris* can grow without any significant inhibition at natural pH values (pH 7 ± 0.2) up to concentrations of 1000 mg-N/L [49]. However, considering the experimental conditions specified in the study, it can be calculated that the FA concentration, which constitutes the main source of ammonia toxicity, is quite low. The inhibitory effects of ammonia on the growth of *C. vulgaris* have been observed at concentrations as low as 10.93 mg-FA/L [14]. In contrast to the study by Kim et al., it was shown that *C. vulgaris* can grow without inhibition at 1600 mg-N/L, which corresponds to 50.49 mg-FA/L, in a very slightly basic (pH range 7.78–7.82) environment [50]. In addition, this study showed that *C. vulgaris* can survive at concentrations as high as 226.1 mg-FA/L. An IC_50_ value as high as 184 mg-FA/L was reported for *C. vulgaris* [13]. According to the results obtained, *C. vulgaris* demonstrated the highest reproductive capability among the four species studied while showing high resistance to ammonia inhibition. According to the substrate inhibition model, *C. vulgaris* was calculated to have the highest theoretical specific growth rate of 1.54 d^−1^. Specifically, *C. vulgaris* exhibited a specific growth rate of 1.243 d⁻¹ at a concentration of 6.77 mg-FA/L, with a doubling time of 13.39 h. Furthermore, substrate inhibition modeling (Figure 1A) indicates that *C. vulgaris* has the potential for faster growth at an optimal FA concentration between 6.77 and 16.93 mg-FA/L.

The IC_50_ value for *C. minutissima* is calculated to be 30.17 mg-FA/L, which corresponds to about 446 mg-
${\mathrm{NH}}_{4}^{+}$
/L. The specific growth rate calculated for *C. minutissima* is 1.44 day^−1^ at the optimum ammonia concentration. The growth and inhibition pattern of *C. minutissima* is similar to *C. vulgaris*, and the calculated specific growth rate and IC_50_ value of *C. minutissima* are slightly lower than *C. vulgaris*. Although there are many studies in the literature showing that *C. minutissima* was grown using ammonia [24,30,51], these studies did not attempt to observe the effect of ammonia toxicity by comparing the effects of different ammonia concentrations. Therefore, the values obtained in this study are considerably higher than other values in the literature. It has been shown that *C. minutissima* can multiply at a concentration of 200 mg-N/L ammonia under autotrophic conditions [30]. In another study, at neutral pH, 250 mg-N/L ammonia concentration did not cause any negative effects on the growth of *C. minutissima* and higher growth was achieved at this concentration than at lower concentrations [31]. 

The lowest calculated IC_50_ value belongs to *C. reinhardtii* and its IC_50_ value was calculated as 27.20 mg-FA/L, which corresponds to about 402 mg-
${\mathrm{NH}}_{4}^{+}$
/L. The response of *C. reinhardtii* to different ammonia concentrations is quite different from the other two Chlorella species tested in this study. Although *C. reinhardtii* has the lowest IC_50_ value, its highest theoretical specific growth rate, calculated using the substrate inhibition model, is 1.53 day⁻¹, which is nearly identical to that of *C. vulgaris*. Moreover, it was observed that *C. reinhardtii* had a higher specific growth rate in media with concentrations of to 6.77 mg-FA/L (Figure 1A,C). In addition to this, at 6.77 mg/L FA concentration, the doubling time of *C. reinhardtii* was calculated to be as low as 11.39 h in one of the sets. This indicates that *C. reinhardtii*’s affinity for ammonia is higher than *C. vulgaris*, but also more susceptible to ammonia toxicity. *C. reinhardtii* can grow at natural pH values (pH 7.2) with concentrations up to about 265 mg-N/L [52]. It has been reported that the inhibitory effects of ammonia begin to manifest at concentrations of 1000 mg-N/L under natural pH conditions (6–7) [53]. On the other hand, the inhibitory effect of ammonia on the growth of *C. reinhardtii* was observed at a concentration of 210 mg-N/L at pH 8 [54]. When these two studies are compared, it can be seen that the amount of FA that causes the actual toxic effect is approximately 4.75 times higher in the study by Jo et al. [54] than in the study by Tevatia et al. [52]. However, it is still significantly lower than the value of 16.93 mg-FA/L obtained in this study.

*A. platensis* was found to be the most resistant species to ammonia toxicity in the study. The IC_50_ value for *A. platensis* was calculated as 44.44 mg-FA/L, which corresponds to about 659.5 mg-TAN/L. Other studies have also shown that *A. platensis* is more resistant to ammonia toxicity than *C. vulgaris* [55,56]. On the other hand, among the four species, *A. platensis* has the lowest specific growth rate under optimum ammonia concentration. The specific growth rate calculated for *A. platensis* is 1.28 day^−1^ at the optimum ammonia concentration. *A. platensis* is the slowest-growing species at ammonia concentrations lower than 6.77 mg-FA/L. However, this situation reverses as the ammonia concentration increases to 33.85 mg-FA/L. It is believed that the reason for the slower growth of *A. platensis* at ammonia concentrations where ammonia toxicity is low, compared to other species tested in the study, is due to *A. platensis* being cultivated at a non-optimal pH value. It has been observed that there is a significant decrease in the growth of *A. platensis* at pH values below 8.5 [57]. However, this effect is considered to be less significant than the toxicity of ammonia. In a study conducted by Markou et. al [9], it was shown that the growth of *A. platensis* at pH 8 and 200 mg-N/L ammonia concentration was relatively unaffected, whereas growth stopped completely at pH 9, which is the optimum pH for *A. platensis* [57]. Again, in this study, although growth was significantly negatively affected, growth was observed at pH 9 and pH 10 at 150 mg-N/L ammonia concentration [9]. On the other hand, Markou et al. stated that in the pH range of 8–10, increasing the ammonia concentration up to 100 mg-N/L has limited effects on growth [9]. The FA concentration at pH 10 was calculated to be approximately 108.8 mg-FA/L, and *A. platensis* showed much higher resistance to ammonia toxicity than the results obtained in our own study. Contrary to the results of Markou et. al. [9], and in accordance with the results obtained from our own study, the inhibition of *A. platensis* growth begins at concentrations of 22.4 mg-FA/L, and they noted that a concentration of 131.5 mg-FA/L is toxic [58,59,60]. According to the results obtained in this study, although *A. platensis* showed the highest resistance to ammonia toxicity, the optimum pH value of *A. platensis* is at the values where ammonia stripping occurs, making *A. platensis* a poor candidate in studies for ammonia recovery in wastewater.

It is not possible to know the strength of ammonia toxicity without knowing or calculating the FA concentration. Moreover, based on the results obtained from the literature on *C. vulgaris* and *A. platensis*, it has been seen that just knowing the FA concentration is not sufficient. It is thought that the difference in the results obtained in studies can be explained by the different experimental conditions in the studies along with the FA concentration. Although ammonia toxicity is primarily due to the concentration of FA, the results obtained in this study, along with the wide range of results in the literature, indicate that secondary factors, such as light intensity and initial inoculum amount, can significantly affect the severity of ammonia toxicity. It is well-established that the primary mechanism impacted by ammonium toxicity is photosynthesis [17]. Consequently, light intensity can influence the toxic effects of high ammonium levels, with toxicity typically being more severe at higher light levels [17,56]. Hillebrand and Sommer [61] reported that 300 µM was toxic to Pseudo-nitzschia multiseries when exposed to 230 μmol photon/m^2^/s, but was not toxic at 25 μmol photon/m^2^/s. Markou and Muylaert et al. [56] investigated the effect of increasing light intensity on the degree of ammonia toxicity and the photosynthetic capabilities of *A. platensis* and *C. vulgaris*. Both species responded similarly to increasing light and FA concentrations, and it was observed that photosynthetic activity decreased with increasing light intensity at an ammonia concentration of 27.2 mg-FA/L. When the light intensity reached 150 µmol/m²/s, photosynthetic activity stopped. Considering this finding, the observation that the light intensity of 280 µmol/m²/s used in our study to cultivate microalgae caused toxic effects at much lower FA concentrations is consistent with the results obtained by Markou and Muylaert [56]. It has been shown how variations in the initial amount of inoculum affect *A. platensis*’s response to ammonia toxicity [9]. In the study conducted by Markou et al., while *A. platensis* could not grow at all when the initial inoculum concentration was 250 mg/L at a concentration of 200 mg-N/L ammonia at pH 9, it could grow when the initial inoculum concentration was increased to 500 mg/L [9]. Although it was stated by Azov and Goldman [62] that ammonia inhibition was independent of the initial biomass concentration, this result obtained by Markou et al. needs further explanation.

Han and Levenspiel categorized six common patterns of growth inhibition by comparing the dimensionless constants and m and n [47]. Utilizing this framework, the observed inhibition aligned with a noncompetitive inhibition model, as the calculated dimensionless constants indicated m > n > 0 across all species in the experiment’s growth inhibition model.

There is limited information available on the self-adaptation mechanisms of microalgae when exposed to unfavourable conditions, such as FA shock [20]. Chen et al. [20] reported that *Chlorella* sp. enhances its DNA repair mechanisms and increases its utilization of phosphate and COD in response to FA shock. In addition, increased activity of genes responsible for the production of subunits of PSI and PSII was observed [20]. On the other hand, extensive research exists on the responses of microalgae to sustained FA inhibition during microalgal growth [13,56,63,64]. Under conditions of ammonia toxicity, the activity of glutamate synthase decreases [64]. PEPCase activity increases, facilitating the rapid incorporation of ammonium into organic compounds to mitigate toxicity [17]. Ammonium can also be eliminated through extrusion/efflux into the surrounding medium [17]. It has been observed that under sustained FA inhibition, microalgae produce more extracellular organic matter to protect themselves [13]. It has also been observed that in the presence of chronic FA, microalgae increase the production of the relevant subunits of PSII to replace the damaged subunits [12]. By using these mechanisms with different effectiveness, microalgae reduce ammonia toxicity. Identifying the specific mechanisms employed by different species and assessing their effectiveness in mitigating ammonia toxicity is beyond the scope of this study.

### 3.2. Effects of Ammonia on Nitrogen Removal

When microalgae are used for nutrient removal from wastewater, it is known that ammonia stripping is one of the most important mechanisms of ammonia removal, especially at high pH values [65]. To accurately quantify the nitrogen incorporated into the biomass and assess the FA concentration and its potential effects due to nitrogen stripping, it is essential to measure the extent of nitrogen stripping. To determine if ammonia stripping occurs under operational conditions and to quantify its magnitude, ammonia stripping experiments were conducted for each varying FA concentration. Ammonia stripping experiments, conducted under the same conditions used for microalgae cultivation, were performed for each FA concentration over a duration of 250 h, without any inoculation. The results indicated that, regardless of FA concentration, the nitrogen loss due to ammonia stripping after 250 h was consistently less than 1% and similar across all FA concentrations tested. Our results were, consistent with the literature [65,66]. Ammonia stripping was independent of ammonia concentration and could be disregarded in the necessary calculations for this study.

The relationship between nitrogen removal and variations in FA concentrations is shown in Figure 2. As the FA concentration increased from 1.69 to 6.77 mg/L, all species exhibited similar results, maintaining high rates of nitrogen removal regardless of the increased FA levels. Except for *C. minutissima*, the other three species achieved nitrogen removal efficiencies between approximately 95% and 99% within this FA range. The highest nitrogen removal efficiency of 99.1% was observed in *A. platensis* grown at an FA concentration of 6.77 mg/L. Although *C. minutissima* also demonstrated high nitrogen removal efficiency, its nitrogen removal rates were slightly lower than the other three species in media with an FA concentration lower than 6.77 mg/L, ranging between approximately 89% and 94%. In the literature, similarly high removal efficiencies have been observed in the absence of ammonia toxicity [9,13,53,67,68,69,70]. It has been reported that *C. vulgaris* achieves 100% nitrogen removal in media with ammonia nitrogen concentrations up to 240 mg-N/L [13]. In the study conducted by Markou, nearly 100% nitrogen removal efficiency was achieved by A. platensis, even at high concentrations of up to 175 mg-FA/L under elevated pH conditions [9]. However, as stated in the study, the majority of this removal (up to 80%) was attributed to nitrogen stripping induced by high pH levels [9]. In another study conducted at a high pH (around 10), it was reported that 96% nitrogen removal was achieved in the diluted wastewater effluent of a petrochemical company containing 56.01 mg-N/L ammonia [70]. In the study conducted by Singh et al., it was shown that *C. minutissima* removed 99% of nitrogen at low FA concentrations, approximately 1.4 mg/L (247.6 mg-N/L ammonia) [67]. In another study, it was reported that *C. minutissima* achieved nearly 100% nitrogen removal at an FA concentration of 5.25 mg/L [68]. On the other hand, *C. reinhardtii* has been demonstrated to achieve 100% removal of 55 mg-N/L ammonia under natural pH conditions [69]. In the study conducted by Su et al., it was reported that *C. reinhardtii* achieved 100% nitrogen removal efficiency at an average FA concentration of approximately 15 mg/L [53].

On the other hand, when the FA amount increases from 6.77 to 67.70, although there is a general trend of decreasing nitrogen removal efficiency with increasing FA amounts in all species, nitrogen removal efficiency varies from species to species. While the other three species achieved similar nitrogen removal efficiency at an FA concentration of 16.93 mg/L, *C. reinhardtii* demonstrated a significantly higher removal efficiency, achieving 74.85% nitrogen removal. As the FA concentration increased, the nitrogen removal efficiency of *C. reinhardtii* became similar to those of other species due to its greater susceptibility to ammonia toxicity. At an FA concentration of 67.70 mg/L, all species exhibited similarly low nitrogen removal efficiency, ranging from approximately 1.5% to 2.5%. The nitrogen removal efficiency of *C. vulgaris* declined rapidly at concentrations exceeding 240 mg-N/L [13]. At an ammonia concentration of 500 mg-N/L, the nitrogen removal efficiency dropped below 40% [13]. In the study conducted by Yuan et al., *A. platensis* cultivated in a fed-batch system completely removed the ammonia at loadings up to approximately 11.5 mg-FA/L per day [71]. At higher loadings, the ammonia was not fully consumed and accumulated in the system [71]. They stated that there was no ammonia stripping in their system [71]. Conversely, insufficient information is available regarding the nitrogen removal efficiencies of *C. minutissima and C. reinhardtii* at FA concentrations as high as those examined in this study. However, a review of the literature indicates that the nitrogen removal efficiencies of *C. vulgaris* and *A. platensis* exhibit a pattern similar to that observed in this study when nitrogen stripping is absent [13,70].

As shown in Figure 3, which presents the biomass amounts obtained at different FA concentrations, it was observed that the biomass concentration at the end of cultivation significantly decreased as the FA concentration increased. Due to the negligible ammonia stripping observed in this study, nitrogen removal primarily occurs through the incorporation of nitrogen into the growing microalgae biomass. Consequently, a decrease in nitrogen removal efficiency was observed, corresponding to the negative impact of increased ammonia toxicity on the biomass growth of microalgae. In addition to this, the nitrogen content of the biomass also affects the nitrogen removal efficiency slightly. This is the reason for the deviations in the relationship between nitrogen removal efficiency and the obtained biomass.

As can be understood from the DBW-FA and N. Removal-FA graphs (Figure 2 and Figure 3), at FA concentrations ranging from 1.69 to 6.77 mg/L, the growth-limiting factor is nitrogen deficiency rather than ammonia toxicity. At these concentrations, ammonia toxicity is either absent or its effect is too minimal to be detected based on the results of this study. At an FA concentration of 16.93 mg/L, cell proliferation generally reaches saturation for all species. Although the highest biomass was obtained at this concentration, the species in the study exhibited constrained biomass production despite the presence of sufficient nutrients in the medium to support further growth. Algae have been documented to use a self-monitoring strategy for population regulation, and it has been observed that biomass does not exceed a certain maximum cell density even when environmental conditions are maintained at levels that enable growth [72]. Several reports suggest that algae may mimic quorum sensing to regulate the self-monitoring of their cell population [73,74,75]. Therefore, it is hypothesized that due to the effect of ammonia toxicity on the species in the experiment, they limit their maximum biomass production by regulating their proliferation at a certain threshold, even though there is excess nitrogen in the environment. However, there is currently no established mechanism to explain this phenomenon. Based on the results obtained in this study, it can be concluded that at an FA concentration of 16.93 mg/L, the positive impact of sufficient nitrogen availability on growth is evident, while concurrently, the detrimental effects of ammonia toxicity, such as the suppression of microalgal growth [20], are also observed. To align with these assumptions, a similar pattern to that observed in this study was also reported by Jiang et al. in their research involving *C. vulgaris* [13]. Consequently, the challenge of identifying the optimal ammonia concentration for maximum growth is evident here.

All species have increasing biomass concentrations with increasing FA levels up to a peak at 16.93 mg/L of FA, after which their biomass concentrations decrease. The ANOVA test showed that there were significant differences in the biomass produced as a result of growth among species (F = 636.34, *p* < 0.05). *C. vulgaris* was the species that produced the highest biomass. The maximum biomass was achieved at an FA concentration of 16.93 mg/L, with the highest biomass amounting to 1700.95 mg/L. Moreover, *C. vulgaris* produced the highest biomass at all FA concentrations, except for the 33.85 mg/L FA concentration. At FA concentrations lower than 16.93 mg-FA/L, *A. platensis* was observed to produce much lower biomass than the other three species. In contrast to *C. vulgaris*, *A. platensis* produced the highest biomass with a biomass amount of 805.11 mg/L at a FA concentration of 33.85 mg/L.

The change in nitrogen concentration over time at different FA concentrations is shown in Figure 4. It has been observed that in media with FA levels below 6.77 mg/L, most of the ammonia is consumed within the first 150 h. This point corresponds to the transition from the exponential growth phase to the stationary phase for all species. As the FA concentration increases, nitrogen cannot be completely consumed, resulting in a gradual decrease in the nitrogen consumption rate. The pattern of decrease in nitrogen concentration is similar for all FA concentrations. For all FA concentrations, the nitrogen consumption rate was found to be high in the exponential growth phase, while it was quite low in the lag and stationary phases, consistent with the growth curve.

The experimental data were used for the determination of N content of biomass, biomass yield, and average nitrogen removal rate. The results obtained are given in Table 3. At the end of the experiment, it was observed that the nitrogen content of the species varied between approximately 4% and 10%. The highest nitrogen content, at 10.25%, was found in *C. reinhardtii* grown at a 33.85 mg/L FA concentration, while the lowest nitrogen content, at 3.98%, was observed in *C. minutissima* grown at a 67.70 mg/L FA concentration. At low FA concentrations, algae species exhibit moderate to high nitrogen content percentages, indicating effective assimilation of nitrogen into their biomass. It is seen that the nitrogen content in the biomass also increases with increasing FA concentrations of up to 6.77 mg/L FA concentration. After this, the nitrogen content remains stable or slightly increases at moderate FA concentrations, suggesting that these conditions might be optimal for nitrogen assimilation. At 50.78 mg/L FA concentration and higher FA concentrations, the nitrogen concentration in biomass decreased significantly, despite the availability of more nitrogen in the environment. In accordance with the results obtained in our study, a similar pattern was observed between the nitrogen concentration in the medium and the protein concentration in the cell in the study conducted by Zarrinmehr [76]. This study demonstrated that the protein content of *Isochrysis galbana* microalgae is influenced by the nitrogen concentration in the medium. Specifically, the protein concentration within the cell increases as the nitrogen concentration in the medium reaches up to 144 mg/L, but decreases rapidly at concentrations exceeding 144 mg/L. On the other hand, in the study conducted by Jiang et al., no correlation was detected between the protein content and FA concentration in *C. vulgaris* [13]. 

Based on the results obtained, excluding *C. minutissima*, microalgae species examined in the experiment exhibited a similar pattern for biomass yield. It has been observed that biomass yield decreases with increasing FA concentration for these species. On the other hand, while it could be said that increasing FA concentration has a somewhat negative impact on the biomass yield of *C. minutissima*, the results obtained are highly variable. Compared to other species, the nitrogen content of *C. minutissima* is much lower at high FA concentrations, causing fluctuations in biomass yield. The highest biomass yield observed for *C. minutissima*, which demonstrated the greatest biomass production, is 19.52 (g-X/g-N_con_). On the other hand, in the study conducted by Malla et al., it was found that the biomass yields of *C. minutissima* grown under low ammonia conditions were greater than 36 (g-X/g-N_con_), which is even higher than the results obtained in our study [68]. Although *C. minutissima* exhibited high biomass yield at all FA concentrations (except 6.77 mg-FA/L), making it a very good candidate for biomass production at low nitrogen concentrations, its nitrogen removal rate and biomass production were lower than those of *C. vulgaris*, rendering *C. minutissima* less successful in the context of this study. At non-toxic ammonia concentrations, the biomass yield of *C. vulgaris* is approximately 17–18 (g-X/g-N_con_). Jiang et. al., reported findings consistent with those of this study, with a biomass yield of approximately 20 (g-X/g-N_con_) [13]. While *C. vulgaris* exhibits a biomass yield close to that of *C. minutissima*, it achieves the highest biomass production at all concentrations except for the 16.93 mg/L FA concentration.

The average nitrogen removal rate data for all species show similarities, with the highest values obtained at moderate FA concentrations. This is thought to be due to the optimal balance of nitrogen availability and manageable levels of stress from FA. All species exhibit similar average nitrogen removal rates in media with FA concentrations up to 6.77 mg/L. In the medium containing 16.93 mg/L FA, the three species, excluding *C. reinhardtii*, exhibited similar average nitrogen removal rates. *C. reinhardtii* has the highest average nitrogen removal rate with 13.95 mg/L*day at 16.93 mg/L FA concentration, which is significantly higher than other species (F = 158.37, *p* < 0.05). In the study conducted by Su et al., similar results were obtained for *C. vulgaris* (3.66 mg/L*day) at low ammonia concentrations as in this study, while significantly better results were achieved for *C. reinhardtii* (6.39 mg/L*day) [53]. Conversely, it was observed that *C. vulgaris* achieved average nitrogen removal rates between 15 and 26.4 mg/L*day at moderate FA concentrations (11–24 mg-FA/L), which is significantly higher than the rates obtained in our study [13]. This discrepancy is primarily attributed to the prolonged stationary phase duration in our study. During the stationary phase, nitrogen consumption is minimal, resulting in a lower average nitrogen removal rate. Although *C. reinhardtii* produces less biomass than *C. vulgaris*, it exhibits the highest average nitrogen removal rate in media with FA concentrations below 33.85 mg/L. This suggests that *C. reinhardtii* possesses a significantly higher nitrogen content (up to 10.25%) compared to other species, resulting in a superior average nitrogen removal rate.

## 4. Conclusions

Microalgae cultivation strategies utilizing wastewater with high ammonia concentration are crucial for reducing the costs associated with nitrogen inputs required to support biomass production. In this study, the effect of ammonia inhibition on four microalgae species frequently studied for wastewater treatment and production of high-value products was observed, and inhibition and nitrogen removal kinetics were calculated for each species. The results showed that all tested species could grow without inhibition at ammonia levels similar to those found in municipal wastewater. Furthermore, it has been concluded that the species *C. vulgaris* and *A. platensis* can tolerate high ammonia levels similar to those found in strong wastewaters. Under the conditions in this experiment, *C. vulgaris* was determined to be the best candidate to be used in the treatment of high ammonia-containing wastewater among the four species studied, based on its resistance to ammonia toxicity, biomass production, and ammonia removal capacity. The IC_50_ values of *C. vulgaris* were calculated as 34.82 mg-FA/L, and the highest biomass production of 1700 mg/L was obtained with *C. vulgaris*. It is believed that the insights provided by this study, which demonstrates the ammonia responses of microalgae, will be applicable in the field of microalgae-based wastewater treatment.

## Figures and Tables

**Figure 1 microorganisms-12-01542-f001:**
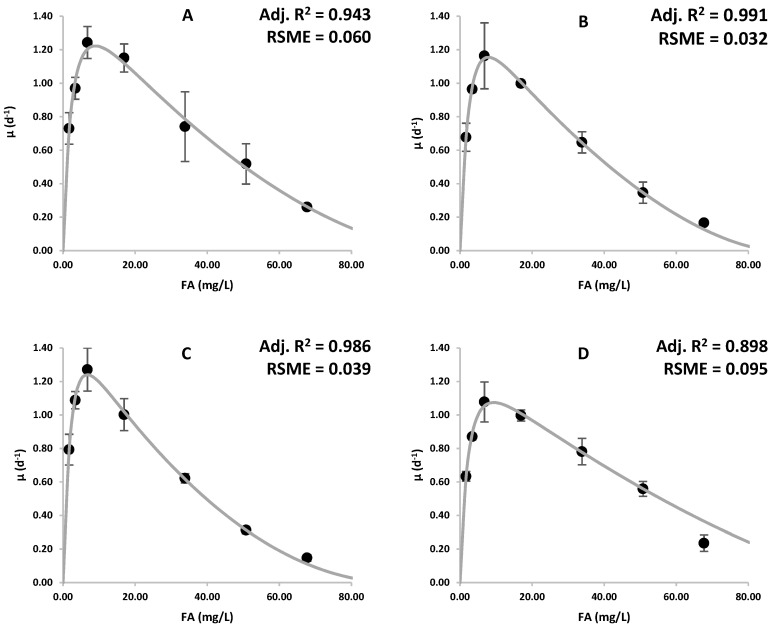
Specific growth rate versus FA curves for each species: (**A**) *C. vulgaris*, (**B**) *C. minutissima* (**C**) *C. reinhardtii* and (**D**) *A. platensis*, represented by circles, reported as d^−1^ over FA concentrations. The error bars indicate the standard deviation of the specific growth rate for each growth rate experimental condition. The output of the substrate inhibition model using Equation (9) is represented by the line. The root mean squared error (RMSE) and Adj. R^2^ were used to evaluate the goodness of fit of the curve fitting.

**Figure 2 microorganisms-12-01542-f002:**
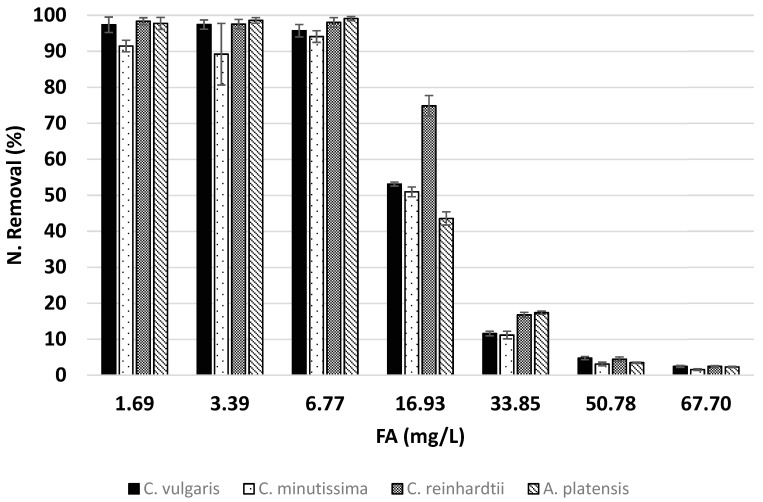
Ammonia removal efficiency. The error bars indicate the standard deviation of the specific growth rate for each growth rate experimental condition.

**Figure 3 microorganisms-12-01542-f003:**
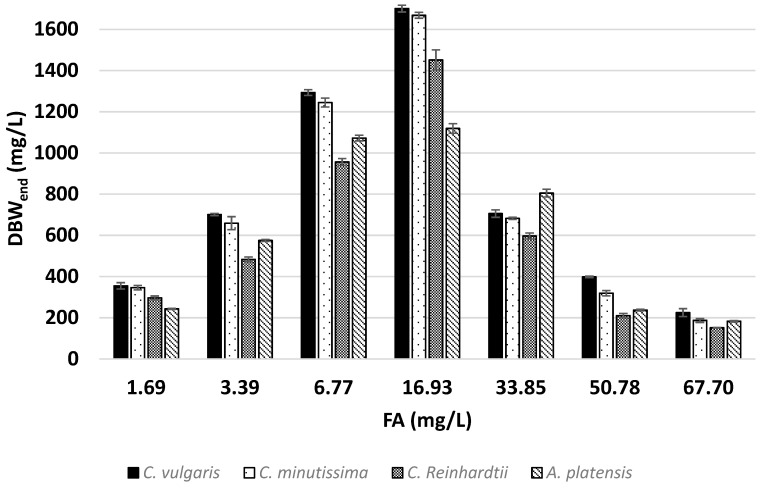
DBW production amount at the end of cultivation. The error bars indicate the standard deviation of the specific growth rate for each growth rate experimental condition.

**Figure 4 microorganisms-12-01542-f004:**
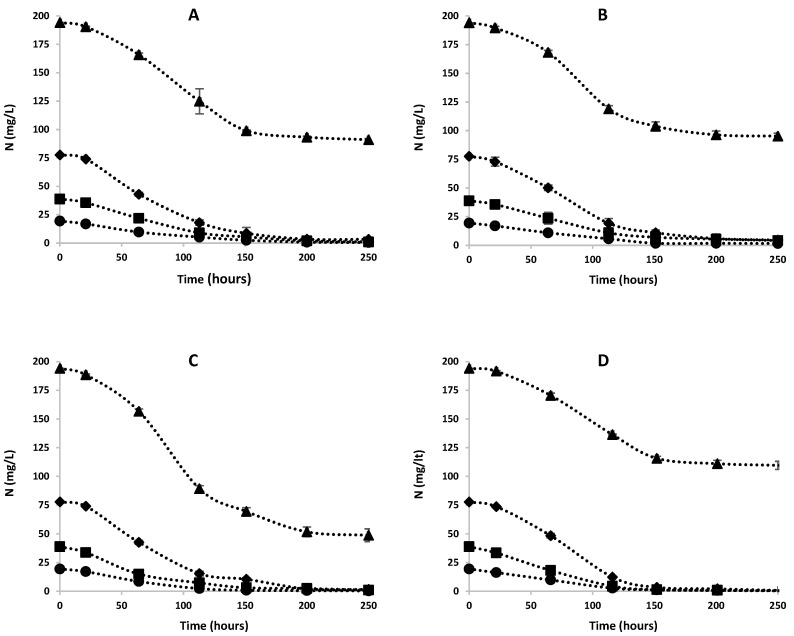
Nitrogen concentration (mg/L) versus time (hours) graph for different FA concentrations for each species: (**A**) *C. vulgaris*, (**B**) *C. minutissima*, (**C**) *C. reinhardtii* and (**D**) *A. platensis*. ●: 1.69 mg-FA/L, ■: 3.39 mg-FA/L, ♦: 6.77 mg-FA/L, ▲: 16.93 mg-FA/L.

**Table 1 microorganisms-12-01542-t001:** Composition of Bold’s Basal Medium. In the experimental cultures, nitrate omitted BBM, containing ammonium chloride as a nitrogen source, was used.

Name	Conc.	
	mM	mg/L
NaNO_3_	2.94	249.88
MgSO_4_.7H_2_O	0.304	74.93
NaCl	0.428	25.01
K_2_HPO_4_	0.431	75.08
KH_2_PO_4_	1.29	175.55
CaCl_2_.2H_2_O	0.17	24.99
ZnSO_4_.7H_2_0	0.031	8.83
MnCl_2_.4H_2_0	7.28 × 10^−3^	1.44
MoO_3_	4.93 × 10^−3^	0.71
CuSO_4_.5H_2_0	6.29 × 10^−3^	1.57
Co(NO_3_)_2_.6H_2_O	1.68 × 10^−3^	0.49
H_3_BO_3_	0.185	11.44
Na_2_EDTA.2H_2_O	0.171	63.65
KOH	0.553	31.03
FeSO_4_.7H_2_O	0.018	4.98
H_2_SO_4_	0.018	1.80

**Table 2 microorganisms-12-01542-t002:** Summary of growth parameters for microalgal growth. μ_max_ (calculated maximum growth rates, day^−1^), K_m_ (Monod half-saturation concentration, mg-FA/L), K_i_ (inhibition concentration, mg-FA/L), IC_50_ (50% growth inhibition concentration, mg-FA/L), and m and n are unitless constants. Goodness of fit of the inhibition model according to Adjusted R^2^: *C. vulgaris*: 0.943, *C. minutissima*: 0.991, *C. reinhardtii*: 0.986 and *A. platensis*: 0.898.

Parameter	*C. vulgaris*	*C. minutissima*	*C. reinhardtii*	*A. platensis*
μ_max_	1.54	1.44	1.53	1.28
K_i_	106.50	87.76	92.52	118.09
K_m_	2.28	2.48	2.23	2.16
n	1.75	1.64	1.99	1.47
m	13.67	16.56	22.75	18.03
IC_50_	34.82	30.17	27.20	44.44

**Table 3 microorganisms-12-01542-t003:** Nitrogen removal parameters for each species grown with different FA concentrations. N Cont.: Nitrogen content of the biomass at end of growth; N Rem. Eff.: Nitrogen Removal Efficiency; Y_X/N_: Amount of biomass produced per amount of nitrogen consumed; Ave. N. Rem. Rate: Average nitrogen removal rate.

Assay		DBW_end_ (mg/L)	N Cont. (%)	N Rem. Eff. (%)	Y_X/N_ (g-X/g-N_con_)	Ave. N Rem. Rate (mg/L*day)
Species	FA Conc. (mg/L)					
*C. vulgaris*	1.69	355.10	5.30	97.36	18.79	1.81
	3.39	700.86	5.37	97.43	18.53	3.63
	6.77	1293.32	5.72	95.71	17.40	7.13
	16.93	1700.95	6.00	53.12	16.50	9.90
	33.85	705.68	5.83	11.59	15.69	3.93
	50.78	398.08	5.33	4.75	14.39	2.23
	67.70	225.20	4.25	2.46	11.78	1.43
*C. minutissima*	1.69	346.63	5.10	91.46	19.52	1.70
	3.39	659.04	5.22	89.19	19.03	3.32
	6.77	1244.62	5.84	94.08	17.04	7.01
	16.93	1668.47	5.87	50.96	16.87	9.50
	33.85	682.70	5.77	11.16	15.75	3.78
	50.78	319.55	4.55	3.11	17.64	1.46
	67.70	187.11	3.98	1.56	15.45	0.91
*C. reinhardtii*	1.69	296.74	6.40	98.34	15.54	1.83
	3.39	483.01	7.80	97.52	12.76	3.63
	6.77	955.93	7.93	98.03	12.56	7.31
	16.93	1451.60	9.94	74.85	9.99	13.95
	33.85	596.96	10.25	16.77	9.17	5.68
	50.78	210.68	9.24	4.47	8.10	1.94
	67.70	151.44	6.30	2.45	7.96	1.33
*A. platensis*	1.69	243.31	7.76	97.75	12.82	1.81
	3.39	574.95	6.66	98.58	15.02	3.66
	6.77	1072.27	7.14	99.10	13.94	7.36
	16.93	1118.93	7.39	43.56	13.23	8.09
	33.85	805.11	7.89	17.36	11.94	5.86
	50.78	237.66	5.85	3.50	11.66	1.64
	67.70	182.80	4.17	2.31	10.21	1.23

## Data Availability

The raw data supporting the conclusions of this article will be made available by the authors on request.

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
