# Peer review of "Evaluating Ammonia Toxicity and Growth Kinetics of Four Different Microalgae Species"

_microorganisms, 2024, doi:10.3390/microorganisms12081542_

Round 1

Reviewer 1 Report

Comments and Suggestions for Authors

Title:

Evaluating Ammonia Toxicity and Growth Kinetics of Four Different Microalgae Species.

Recommendation:

    Major revision.

Comments:

This manuscript evaluates the impact of ammonia toxicity on various microalgae species and their growth kinetics in the context of wastewater treatment and biofuel production. It explores the tolerance levels of different microalgae strains to high concentrations of ammonia, focusing on species like Chlorella vulgaris, Chlorella minutissima, Chlymodomanas reinhardtii, and Arthrospira platensis. The study aims to understand the potential of these microorganisms in efficiently removing nitrogen from wastewater and their suitability for biofuel production in ammonia-rich environments. The subject is relevant and consistent with the aims and scopes of the journal. The subject is relevant and consistent with the aims and scopes of the journal. Several comments and suggestions are offered below with the intent to assist the author in improving the manuscript.

1.      The author's affiliation information is incomplete. Please revise it carefully.

2.      There are numerous formatting errors in the manuscript. For example, the references are incorrectly cited in the manuscript, and section headings should not be enlarged. Please revise these issues according to the "Author Guidelines."

3.      In some parts of the manuscript, when referring to the microalgae C. minutissima, the species name's first letter should not be capitalized. For instance, in Table 2, it is written as C. Minutissima, and in Table 3, the species names of all algae are capitalized. Please revise these carefully.

4.      The title of Table 1 should be detailed, or consider removing Table 1 and including the medium composition within the manuscript.

5.      On lines 390 and 391, the font size increases. Please correct this.

6.      References 35 and 36 are duplicates.

7.      What are the potential mechanisms underlying the varying levels of resistance to ammonia toxicity among the different microalgae species studied?

8.      How can the insights gained from this research be applied to optimize microalgae cultivation in real-world wastewater treatment systems to enhance efficiency and sustainability?

Comments on the Quality of English Language

Minor editing of English language required

Author Response

Comments 1: [The author's affiliation information is incomplete. Please revise it carefully.]

Response 1: [The author's affiliation information was completed.] Thank you for pointing this out. [The missing membership information of the authors was completed and the necessary corrections were made. The author names on page 1, line 4 have been corrected to " Umut Metin 1 and Mahmut Altınbaş 2,*". On page 1, the author information has been changed to the following:

1    Department of Environmental Engineering, Faculty of Civil Engineering, Yıldız Technical University, 34220 Istanbul, Turkey; umetin@yildiz.edu.tr

2    Department of Environmental Engineering, Faculty of Civil Engineering, Istanbul Technical University, 34469 Istanbul, Turkey; altinbasm1@itu.edu.tr

*    Correspondence: altinbasm1@itu.edu.tr

]

Comments 2: [There are numerous formatting errors in the manuscript. For example, the references are incorrectly cited in the manuscript, and section headings should not be enlarged. Please revise these issues according to the "Author Guidelines.".]

Response 2: We have revised the article to emphasize this point. [Format of this article has been arranged based on the Microorganisms Microsoft Word template file. The following changes have been made to correct formatting errors:

  • On page 1, the Keyword section has been added.
  • The format of all titles has been changed to Palatino Linotype font, size 10.
  • Equations have been adapted to the format.
  • The font for the entire article is set to "Palatino Linotype". Font sizes have been adjusted as shown in the template.
  • The References has been edited in accordance with the format.
  • References are numbered in order of appearance in the text.
  • The references in the article are provided with reference numbers in square brackets, in accordance with the required format.
  • Added Disclaimer/Publisher's Note

]

Comments 3: [In some parts of the manuscript, when referring to the microalgae C. minutissima, the species name's first letter should not be capitalized. For instance, in Table 2, it is written as C. Minutissima, and in Table 3, the species names of all algae are capitalized. Please revise these carefully.]

Response 3: Thank you for pointing this out. [As you highlighted, errors related to the writing of species names have been reviewed. The following corrections have been implemented:

  • On page 2, in line 71, the font of the word "Arthrospira" has been changed to Palatino Linotype, and the font size has been set to 10.
  • In page 7, the species names mentioned in lines 268 and 269 have been converted to italic format.
  • On page 8, in Table 2, the word "Minutissima," which started with a capital letter, has been changed to begin with a lowercase letter.
  • On page 8, the word vulgaris in line 292 has been converted to italic format.
  • On page 8, the word vulgaris in line 313 has been converted to italic format. Additionally, the word "Chlorella" has been abbreviated to "C."
  • On page 8, the word vulgaris in linse 316 and 318 has been converted to italic format.
  • On page 9, the word arthrospira in line 367 has been converted to italic format.
  • On page 13, in Figure 4, the word "Minutissima," which started with a capital letter, has been changed to begin with a lowercase letter. Additionally, the formatting of species names
  • in Figure 4 has been changed to italic.
  • On page 13, in line 508, the word "Minutissima," which started with a capital letter, has been changed to begin with a lowercase letter.
  • On page 14, in Table 3, the words "Vulgaris," "Minutissima," "Reinhardtii," and "Platensis," which started with capital letters, have been changed to begin with lowercase letters. Additionally, the formatting of species names in Table 3 has been changed to italic.
  • On page 14, the word minutissima in lines 526, 527, 528 and 531 has been converted to italic format.
  • On page 15, the word reinhardtii in lines 537, 538, 540 and 542 has been converted to italic format. On page 15, the word vulgaris in line 553 has been converted to italic format.]

Comments 4: [The title of Table 1 should be detailed, or consider removing Table 1 and including the medium composition within the manuscript.]

Response 4: We have modified to title of table 1 to emphasize this point [The title of Table 1 was changed to “Composition of Bold’s Basal Medium. In the experimental cultures, nitrate-omitted BBM, containing ammonium chloride as a nitrogen source, was used.” Changes were made on page 2.]

Comments 5: [On lines 390 and 391, the font size increases. Please correct this.]

Response 5: [Corrected the font size on lines 390 and 391.] Thank you for pointing this out. [Corrected the font size on lines 390 and 391. The faulty font was changed to "Palatino Linotype". Changes were made on page 10, lines 390 and 391.]

Comments 6: [References 35 and 36 are duplicates.]

Response 6: [One of the two duplicate references was deleted.] Thank you for pointing this out. [The change was made on page 16.]

Comments 7: [What are the potential mechanisms underlying the varying levels of resistance to ammonia toxicity among the different microalgae species studied?]

Response 7: It's being worked on.

Comments 8: [How can the insights gained from this research be applied to optimize microalgae cultivation in real-world wastewater treatment systems to enhance efficiency and sustainability?]

Response 8: It's being worked on.

Reviewer 2 Report

Comments and Suggestions for Authors

The document do not comply with MDPI format for references. please comply with the guidelines

the document have at least 3 types of fonts. please maintain homogeneity using palatino lynotype font

Line 21. algae are not as efficient as some want to show. please remove that statement

statement in lines 30-32 requires citation

the final idea from line 75-78 is redundant and gives no importance to the literature reviewed.

line 81. change "Pure cultures of" to "axenic unialgal cultures of" 

please add the codes or reference numbers for each strain used.

lines 263. revise the writing "furthermorehis". also revise line 433

please add a proper statistical analysis (maybe a two-way ANOVA) to be sure which strain on which FA concentration produces more biomass and removes more N

Author Response

Comments 1: [The document do not comply with MDPI format for references. please comply with the guidelines.]

Response 1: [Necessary arrangements were made.] We have revised the article to emphasize this point. [The format of the references has been adjusted based on the Microorganisms Microsoft Word template. The following corrections have been made to address formatting errors:

  • The References has been edited in accordance with the format (exp: “Author 1, A.B.; Author 2, C.D. Title of the article. Abbreviated Journal Name Year, Volume, page range.”)
  • References are numbered in order of appearance in the text.
  • The references in the article are provided with reference numbers in square brackets, in accordance with the required format. 

Comments 2: [the document have at least 3 types of fonts. please maintain homogeneity using palatino lynotype font]

Response 2: Thank you for pointing this out. [The font for the entire article is set to "Palatino Linotype". Font sizes have been adjusted as shown in the template. (Note: The mathematical equations added to the article have also been converted to Palatino Linotype font. However, although the text within these equations is in Palatino Linotype format, the spaces within the equations still appear in the default Cambria Math format. I am unable to change this in Word, but it is not noticeable in the appearance.)]

Comments 3: [Line 21. algae are not as efficient as some want to show. please remove that statement]

Response 3: As you pointed out, the relevant sentence has been revised to avoid any scientific misinterpretation. [The word efficient in the relevant sentence was removed from the sentence.

The new sentence has been revised to "Microalgae can function as green chemical factories that can convert sunlight, carbon dioxide, and nutrients into various valuable compounds.". The change can be seen on page 1, line 25.]

Comments 4: [statement in lines 30-32 requires citation]

Response 4: It's being worked on.

Comments 5: [the final idea from line 75-78 is redundant and gives no importance to the literature reviewed.]

Response 5: It's being worked on.

Comments 6: [line 81. change "Pure cultures of" to "axenic unialgal cultures of"]

Response 6: The change you specified has been made. [The new sentence has been revised to " Axenic unialgal cultures of Chlorella vulgaris (CCAP 211/116) and Arthrospira platensis (SAG 85.79) species were purchased from Phycotec Biyoteknoloji A.Ş., while, Chlorella minutissima (ELSTER 1998/9) and Chlamydomonas reinhardtii (SMITH 1980/UTEX 2246) species were obtained from Culture Collection of Autotrophic Organisms (CCALA).”. The change can be seen on page 2, line 82.]

Comments 7: [please add the codes or reference numbers for each strain used.

Response 7: [Relevant codes/reference numbers added.] The change you specified has been made [The new sentence has been revised to " Axenic unialgal cultures of Chlorella vulgaris (CCAP 211/116) and Arthrospira platensis (SAG 85.79) species were purchased from Phycotec Biyoteknoloji A.Ş., while, Chlorella minutissima (ELSTER 1998/9) and Chlamydomonas reinhardtii (SMITH 1980/UTEX 2246) species were obtained from Culture Collection of Autotrophic Organisms (CCALA).”. The change can be seen on page 2, line 82.]

Comments 8: [lines 263. revise the writing "furthermorehis". also revise line 433]

Response 8: Thank you for pointing this out. [The " Furthermorehis effect is thought to" part was deleted from the relevant sentence and "Morever" was written instead. The change was made on page 7, line 253. In line 433, the relevant sentences were rearranged to make the text more understandable. The new sentences have been revised to “The highest nitrogen removal efficiency of 99.1% was observed in A. platensis grown at an FA concentration of 6.77 mg/L. Although C. minutissima also demon-strated high nitrogen removal efficiency, its nitrogen removal rates were slightly lower than the other three species in media with an FA concentration lower than 6.77 mg/L, ranging between approximately 89% and 94%.”. The change can be seen on page 11, between lines 411 and 419]

Comments 9: [please add a proper statistical analysis (maybe a two-way ANOVA) to be sure which strain on which FA concentration produces more biomass and removes more N]

Response 9: The change you specified has been made [As you suggested, the accuracy of our previous results was confirmed by comparing the biomass production and average nitrogen removal rates of the species growing at 16.93 mg/L FA concentration, which is where the highest biomass production and average nitrogen removal rate are.  Necessary arrangements were made under the heading "2.7. Statistics". Changes can be seen on page 6, between lines 225 and 228, page 13, between lines 466 and 468 and page 15, line 536]

Reviewer 3 Report

Comments and Suggestions for Authors

Dear Authors,

In this article, the authors study the effect of different concentrations of ammonium on 4 different strains of microalgae. Apart from the points I am indicating to them, a key point the authors must improve is the discussion; a large part of the data is not discussed in relation to the existing literature. Importantly, studies on ammonium decontamination from wastewater are not analyzed. The authors should compare their data with such studies, as they claim their study's data would contribute to that

Majors:

*L16: ” It has been concluded that species C. vulgaris and Arthrospira platensis can tolerate the high ammonia levels found in strong wastewaters, such as digestate. Additionally, all tested species can tolerate the ammonia levels present in municipal wastewaters.Eliminate,' it's confusing; it seems to indicate that the authors have studied wastewater, but that's not the case

* In the abstract, introduction and conclusions, reference is made to the importance of this study in wastewater decontamination. However, throughout the article, there is no discussion of related studies, nor are the results contextualized within such works. It is important for the authors to discuss the significance of their findings in relation to these studies

*L63: While….species. Please include the reference

*In the introduction, I noticed a lack of citations for studies that use these algae to remove ammonium from wastewater. If such studies have been conducted, please mention them

*L70: Cite based on which studies you have decided to use these 4 species

* Add the statistical error to the data in Table 2ç

* The way of citing by placing the year at the end of the sentence is not correct

-L427-L447: These data are not discussed in relation to the existing literature

-L461-L477:  These data are not discussed enough in relation to the existing literature

- The data in Fig. 4 do not have statistical error

-L515: “contrary to our expectations.” Better justify the reason for this idea

- Throughout the entire text, the names of the 4 species are sometimes italicized and sometimes not

-L521-L542: These data are not discussed enough in relation to the existing literature

-L541: “ This high value is due to C. reinhardtii's ability to assimilate  nitrogen into its biomass at a high rate.” On what bibliographic citation is this statement based

Minors:

L71: Arthrospira. Type

"In Table 1, 'Con. (per L)' doesn't make sense if 'mM' is indicated below

-Table 1: CaCl2*2H2O?

-Throughout different parts of the text, such as in the materials and methods section, different font sizes can be observed

-L110: Cao et. Al. cite appropriately

L284: “conformed”? Please choose a better synonym.

L289: “put aside”? Please choose a better synonym

L294: “e. (P” Type

L296: “Collos and Harrision, “ cite appropriately

L298: “Przytocka-Jusiak” cite appropriately

L297: “3 μM. (2014).” Type

L332: “s. (Arora et. al., 2016).” Type

L362: “8.5. (Ismaiel” Type

L364: “Markou et. Al” cite appropriately

L471: “s. (Taheri and Shariati, 2013).” Type

Author Response

Comments 1: [L16: ” It has been concluded that species C. vulgaris and Arthrospira platensis can tolerate the high ammonia levels found in strong wastewaters, such as digestate. Additionally, all tested species can tolerate the ammonia levels present in municipal wastewaters.” Eliminate,' it's confusing; it seems to indicate that the authors have studied wastewater, but that's not the case]

Response 1: Thank you for pointing this out. [The relevant sentences have been revised to avoid the mentioned misunderstanding. New sentences have been revised to “The results showed that all tested species could tolerate ammonia levels comparable to those found in municipal wastewater. Furthermore, it has been concluded that species C. vulgaris and A. platensis can tolerate high ammonia levels similar to those found in strong wastewaters, such as digestate.”. The change can be seen on page 1, between lines 18 and 21]

Comments 2: [In the abstract, introduction and conclusions, reference is made to the importance of this study in wastewater decontamination. However, throughout the article, there is no discussion of related studies, nor are the results contextualized within such works. It is important for the authors to discuss the significance of their findings in relation to these studies]

Response 2: It's being worked on.

Comments 3: L63: While….species. Please include the reference

Response 3: It's being worked on.

Comments 4: In the introduction, I noticed a lack of citations for studies that use these algae to remove ammonium from wastewater. If such studies have been conducted, please mention them

Response 4 It's being worked on.

Comments 5: L70: Cite based on which studies you have decided to use these 4 species

Response 5: It's being worked on.

Comments 6: Add the statistical error to the data in Table 2ç

Response 6: It's being worked on.

Comments 7: [The way of citing by placing the year at the end of the sentence is not correct]

Response 7: Thank you for pointing this out. [The errors you mentioned were corrected while changing the citation format of the article.]

Comments 8: L427-L447: These data are not discussed in relation to the existing literature

Response 8: It's being worked on.

Comments 9: L461-L477:  These data are not discussed enough in relation to the existing literature

Response 9: It's being worked on.

Comments 10: [The data in Fig. 4 do not have statistical error]

Response 10: Thank you for pointing this out. [Standard deviation data was added to Figure 4 as you requested. However, But I should point out that the error bars are quite difficult to see.]

Comments 11: [L515: “contrary to our expectations.” Better justify the reason for this idea]

Response 11: Thank you for pointing this out. [The assumption I made here without looking at the literature caused an error. After your notice, we found that there were similar results in the literature to the results we obtained in this study. We rewrote the relevant part according to the information we had. The rewritten part can be seen on page 14, lines 500 to 508.]

Comments 12: [Throughout the entire text, the names of the 4 species are sometimes italicized and sometimes not]

Response 12: As per your request, the necessary data has been included in the article. [All species names have been corrected to be written in italics throughout the article.]

Comments 13: [L521-L542: These data are not discussed enough in relation to the existing literature]

Response 13: As per your request, the necessary data has been included in the article. [All species names have been corrected to be written in italics throughout the article.]

Comments 14: [L541: “ This high value is due to C. reinhardtii's ability to assimilate  nitrogen into its biomass at a high rate.” On what bibliographic citation is this statement based]

Response 14: Thank you for pointing this out. [I couldn't explain exactly what I was thinking here. We did not comment here based on the information we obtained from the literature. We made this comment based on the data obtained as a result of our study. Our aim was to explain why C. reinhardtii, despite producing less biomass than C. vulgaris, has the highest average nitrogen removal rate in media with FA concentrations less than 33.85 mg/L. We thought that the reason for this was that the biomass of C. reinhardtii contained a very high amount of nitrogen compared to other species, according to our results (Table 3). This section has been rewritten as “Although C. reinhardtii produces less biomass than C. vulgaris, it exhibits the highest average nitrogen removal rate in media with FA concentrations below 33.85 mg/L. This suggests that C. reinhardtii possesses a significantly higher nitrogen content (up to 10.25%) compared to other species, resulting in a superior average nitrogen removal rate"]

Comments 15: [L71: Arthrospira. Type]

Response 15: Thank you for pointing this out. [The error has been corrected. The change can be seen on page 2, line 75.]

Comments 16: [In Table 1, 'Con. (per L)' doesn't make sense if 'mM' is indicated below]

Response 16: The change you specified has been made [The per L statement has been removed. In the line below, the expression mg is converted to mg/L. The change can be seen on page 3, in table 1]

Comments 17: [CaCl2*2H2O?]

Response 17: Thank you for pointing this out [CaCl2*2H2O expression was corrected to CaCl2.2H2O. The change can be seen on page 3, in table 1]

Comments 18: [Throughout different parts of the text, such as in the materials and methods section, different font sizes can be observed]

Response 18: Thank you for pointing this out. [Format of this article has been arranged based on the Microorganisms Microsoft Word template file. The font for the entire article is set to "Palatino Linotype". Font sizes have been adjusted as shown in the template.]

Comments 19: [L110: Cao et. Al. cite appropriately; L296: “Collos and Harrision, “ cite appropriately; L298: “Przytocka-Jusiak” cite appropriately; L364: “Markou et. Al” cite appropriately]

Response 19: Thank you for pointing this out. [The mentioned sources are appropriately cited. The changes can be seen on page 3, line 119; page 8, line 287; page 9, line 289; and page 10, line 352, respectively.]

Comments 20: [L284: “conformed”? Please choose a better synonym; L289: “put aside”? Please choose a better synonym]

Response 20: Changes you specified has been made [The sentence "The obtained results showed that the growth of the examined microalgae conformed to the substrate inhibition pattern." has been changed to "The obtained results showed that the growth of the examined microalgae is in accordance with the substrate inhibition pattern.". The change can be seen on page 8, in line 277. The word "excluded" was used instead of "put aside". The change can be seen on page 8, in line 281.]

Comments 21: [L294: “e. (P” Type; L297: “3 μM. (2014).” Type; L332: “s. (Arora et. al., 2016).” Type; L362: “8.5. (Ismaiel” Type; L471: “s. (Taheri and Shariati, 2013).” Type]

Response 21: Thank you for pointing this out.  [“3μM. (2014).” typo corrected. The change can be seen on page 8, in line 288. Typographical errors in “e. (P “, "s. (Arora et. al., 2016).", "8.5. (Ismaiel" and s. (Taheri and Shariati, 2013) were corrected while changing the citation format in the relevant sections.]

Round 2

Reviewer 1 Report

Comments and Suggestions for Authors

Title:

Evaluating Ammonia Toxicity and Growth Kinetics of Four Different Microalgae Species

Recommendation:

Accept in current form

Comments:

After the previous review, the manuscript has been revised and modified quite well. Therefore, I suggest that this manuscript can be accepted in current form.

Reviewer 3 Report

Comments and Suggestions for Authors

Dear Authors,

I believe the authors have adequately addressed all of my comments and suggestions, and I accept the paper in its current version.